# Modulatory Role of microRNAs in Triple Negative Breast Cancer with Basal-Like Phenotype

**DOI:** 10.3390/cancers12113298

**Published:** 2020-11-07

**Authors:** Andrea Angius, Paolo Cossu-Rocca, Caterina Arru, Maria Rosaria Muroni, Vincenzo Rallo, Ciriaco Carru, Paolo Uva, Giovanna Pira, Sandra Orrù, Maria Rosaria De Miglio

**Affiliations:** 1Institute of Genetic and Biomedical Research (IRGB), CNR, Cittadella Universitaria di Cagliari, 09042 Monserrato, Italy; vincenzo.rallo@irgb.cnr.it; 2Department of Medical, Surgical and Experimental Sciences, University of Sassari, Via P. Manzella, 4, 07100 Sassari, Italy; rocco@uniss.it (P.C.-R.); mrmuroni@uniss.it (M.R.M.); 3Department of Diagnostic Services, “Giovanni Paolo II” Hospital, ASSL Olbia-ATS Sardegna, 07026 Olbia, Italy; 4Department of Biomedical Sciences, University of Sassari, 07100 Sassari, Italy; cate.cate91@gmail.com (C.A.); carru@uniss.it (C.C.); pira@uniss.it (G.P.); 5CRS4, Science and Technology Park Polaris, Piscina Manna, 09010 Pula, CA, Italy; paolouva@gaslini.org; 6Department of Pathology, “A. Businco” Oncologic Hospital, ASL Cagliari, 09121 Cagliari, Italy; sandra.orru@aob.it

**Keywords:** triple negative breast cancer, basal-like breast cancer, microRNA, epigenetic modulation, TNBC molecular classification, intrinsic molecular subtypes, breast cancer

## Abstract

**Simple Summary:**

Triple Negative breast cancer (TNBC) is an aggressive tumor showing high histological grade, high recurrence, and frequent metastasis, accounting for about 25% of breast cancer-related deaths. Emerging roles and molecular mechanisms by which miRNAs impact pathogenesis and prognosis of basal-phenotype TNBC and their potential clinical utility are analyzed in the present review. Progress achieved in TNBC molecular taxonomy had minimal clinical impact, while miRNAs could act as prognostic/predictive biomarkers for TNBC subtypes. As there are currently no other therapeutic options to treat TNBC apart from chemotherapy, various studies were reviewed to draw the conclusion that ncRNAs might be candidates for drug development and drug resistance. Targeted approaches to epigenetic mechanisms and clarification of the molecular mechanisms of specific miRNAs in TNBC subtypes are fully justified. This review might provide a collection of biomarkers potentially useful in clinical settings and shows that the combination of miRNA-based therapeutic strategies with conventional therapies might synergize anticancer effects improving patient outcome.

**Abstract:**

Development of new research, classification, and therapeutic options are urgently required due to the fact that TNBC is a heterogeneous malignancy. The expression of high molecular weight cytokeratins identifies a biologically and clinically distinct subgroup of TNBCs with a basal-like phenotype, representing about 75% of TNBCs, while the remaining 25% includes all other intrinsic subtypes. The triple negative phenotype in basal-like breast cancer (BLBC) makes it unresponsive to endocrine therapy, i.e., tamoxifen, aromatase inhibitors, and/or anti-HER2-targeted therapies; for this reason, only chemotherapy can be considered an approach available for systemic treatment even if it shows poor prognosis. Therefore, treatment for these subgroups of patients is a strong challenge for oncologists due to disease heterogeneity and the absence of unambiguous molecular targets. Dysregulation of the cellular miRNAome has been related to huge cellular process deregulations underlying human malignancy. Consequently, epigenetics is a field of great promise in cancer research. Increasing evidence suggests that specific miRNA clusters/signatures might be of clinical utility in TNBCs with basal-like phenotype. The epigenetic mechanisms behind tumorigenesis enable progress in the treatment, diagnosis, and prevention of cancer. This review intends to summarize the epigenetic findings related to miRNAome in TNBCs with basal-like phenotype.

## 1. Introduction

Ten to twenty percent of invasive breast cancer (BC) belong to the TNBC type, which is prevalent in young women <50 years of age; among African, American, and Hispanic women; and in women with higher premenopausal body mass index, earlier age at menarche, and higher parity. Specifically, the Triple Negative phenotype (TNP) is defined by ER/PR/HER2-negative immunostaining, habitually with higher expression of the Ki-67 antigen, higher mitotic index, and BRCA1 gene mutations (about 75%). TNBC shows aggressive behavior with high histological grade, high recurrence rate in about 2 or 3 years after treatment compared with other BC subtypes, and frequent distant metastasize; thus, it accounts for about 25% of BC-related deaths [1,2,3,4,5]. TNBC shows different molecular and clinicopathological features [6] and is histologically categorized as a high-grade invasive BC of no special type. “Special types” are still included into the TNBC subtype but differ in biological behavior and clinical course [7].

BLBC is characterized by gene expression usually found in basal or myoepithelial mammary cells, shown by high molecular weight cytokeratin, prevalently CK5/6, and by EGFR expression, with about 75% of them referred to as TNP, with ER/PR/HER2 also being negative by immunohistochemistry (IHC). The remaining 25% includes all other “intrinsic” BC molecular subtypes. BLBC shows short survival following progression to metastatic disease, with prevalence of cerebral and lung metastases with respect to luminal subtypes [8,9]. Remarkably, 80% of TNBCs show basal-like features: TNBC and BLBC phenotypes are effectively synonymous [10], although immunohistochemical expression, transcriptomic, and clinical data suggest they are not equivalent [11,12].

The lack of hormonal and HER2 receptors makes TNBC unresponsive to endocrine therapy, i.e., tamoxifen, aromatase inhibitors, and/or anti-HER2-targeted therapies, so surgery, radiotherapy, and mostly nonspecific chemotherapies (e.g., anthracycline and taxane regimens) remain the mainstay for management of these patients, often with severe side effects affecting life quality [5,13]. TNBC treatment is a major challenge for oncologists, both due to heterogeneity of disease and to the absence of unambiguous molecular targets.

MicroRNAs (miRNAs) are a family of endogenous, short single-stranded, noncoding RNA regulating gene expression by interacting with complementary mRNA-target sequences causing either mRNA degradation or translational repression [14]. MiRNAs regulate multiple processes [14,15], and their dysregulation is strongly related to cancer involving alteration of biological functions [16]. MiRNA level variations were analyzed comparing normal vs. neoplastic tissues [17,18] in several BC subtypes [19] with different responses to endocrine therapy [20]. Several miRNAs were related to pathogenesis of TNBC [21,22] and might represent potential predictors of anticancer drugs efficacy and prognosis [23,24,25]. Recognizing gene expression regulator-classified miRNAs as epigenetic elements involved in cancer development means that they might be potentially used as therapeutic targets and diagnostic/prognostic biomarkers in order to achieve high accuracy in tumor classification [26].

This review summarizes the emerging roles of miRNAs in pathogenesis and prognosis of TNBC with basal-like phenotype (BLP) and discusses their potential clinical utility.

## 2. Molecular Classifications of TNBC: Clinical Outcome Implications

TNBC morphological and molecular heterogeneity, poor prognosis, and lack of specific targeted therapies require a detailed understanding of biology and classification in this type of cancer. Gene expression profiling based on high-throughput technologies provided the basis for an improved TNBC molecular taxonomy, establishing prognostic and predictive indicators. The adoption of unsupervised clustering analysis led to identification of several TNBC molecular subtypes, showing transcriptomic heterogeneity and unique biological pathways.

Lehmann et al. showed that TNBCs include different molecular subtypes, i.e., basal-like 1-2 (BL1-2), immunomodulatory (IM), claudin-low-enriched mesenchymal (M), mesenchymal stem-like (MSL), and luminal androgen receptor (LAR), each one showing a distinctive biology and drug sensitivities [27]. Gene ontology (GO) for the BL-1 subtype is enriched in cell-cycle, cell-division, and DNA damage response (ATR/BRCA) pathways, confirming increased proliferation and loss of cell-cycle checkpoints. BL-2 GO is improved in growth factor receptors (EGFR, MET, and EPHA2) showing specific characteristics of basal/myoepithelial origin with high expression of TP63 and MME mRNAs as well as in glycolysis and gluconeogenesis. The IM subtype is enhanced in immune cell processes (cell and cytokine signaling, antigen processing, and presentation) and signaling by transduction of the immune signal of the nucleus. IM GO overlaps with a gene signature for medullary BC, an unusual, high-grade histologic TNBC subtype showing favorable prognosis [7,28]. The M and MSL subtype GOs are enriched in cell motility, extracellular matrix (ECM) receptor interaction, and cell differentiation. Additionally, the MSL subtype is involved in cell growth and angiogenesis while expresses low levels of proliferation genes with expression enrichment in stem cells, HOX genes, and mesenchymal stem cell-specific markers. The pathway components in the M and MSL subtypes accounts for highly dedifferentiated metaplastic BC, featuring mesenchymal/sarcomatoid or squamous characteristics and chemoresistance [28]. The MSL subtype showed low levels of claudins 3, 4, and 7, congruent with the BC claudin-low subtype [29]. Claudin-low-expressing tumors exhibit a high expression of genes associated with epithelial–mesenchymal transition (EMT). The LAR subtype, although being ER negative, shows a luminal-like expression profile, with a strong expression of AR and downstream AR targets and coactivators. GO is enriched in hormonal-regulated pathways inclusive of steroid synthesis, porphyrin metabolism, and androgen/estrogen metabolism. The LAR subtype consists of AR-driven tumors that encompass the molecular apocrine type.

BC classification as a luminal or basal-like subtype is based on typical protein expression. Luminal BC subtypes express the protein of luminal epithelial cells, called the ‘‘luminal group”, such as luminal cytokeratins (CK8 and 18), ER and GATA3; BLBCs express high molecular weight basal cytokeratins (CK5/6, 14, and 17) and EGFR and/or c-KIT [30]. In TNBC, the BL1-2 and M subtypes express high levels of basal cytokeratin, while the LAR subtype expresses high levels of luminal cytokeratins and luminal markers (FOXA1 and XBP1). To complete the previous BC classification, the intrinsic molecular subtype of BC [31] compared to TNBC subtypes [27] established that 49% of TNBC can be classified as basal-like. A strong association was identified between the BLBC subtype and BL-1 TNBC subtypes, while the BL2, IM, and M subtypes were moderately related to the basal-like molecular class [27]. Finally, a total of 82% of LAR subtypes was classified as Luminal A or B and none were classified as basal-like, strengthening the luminal origin.

Prat et al. showed higher concordance between the TNBC and BLBC subtypes on which was applied the PAM50 intrinsic subtype classifier, finding 78.6% BLP, 7.8% HER2-enriched, 6.6% luminal, and 7% normal-like phenotypes in accordance with three wide clinical trials reviewed by IHC-based and PAM50-based data. Contrariwise, in BLBC, 68.5% was ER−/HER2−, 18.2% was ER+/HER2−, 10.6% was ER−/HER2+, and 2.7% was ER+/HER2+ [12]. The expression profiles in the luminal, HER2-enriched, and basal-like subtypes revealed six main gene clusters [12]. Unsurprisingly, Triple Negative (TN)/luminal exhibited high levels of estrogen-related and luminal genes and low levels of cell cycle-related genes. TN/basal-like included basal epithelial cell and proliferation genes. TN/HER2+ showed increased expression of genes related to oxidation reduction-related biological activities. Moreover, a subcluster of luminal-like genes involving AR was also identified in TN/luminal and TN/HER2+. Although TNBC is biologically heterogeneous, the expression profile of intrinsic molecular subtypes is preserved and no strong differences were observed between intrinsic molecular subtypes with and without TNP [12].

Using the fuzzy clustering method, an unsupervised analysis of gene expression profiles identified three TNBC clusters: C1 (22.4%), C2 (44.9%), and C3 (32.7%). C1 represented a no basal-like phenotype, enriched in luminal and AR genes. C2 was considered pure basal-like. C3 was enriched in BLP including 26% of claudin-low subtypes, and M2-like macrophages were a hallmark of C3 [32].

Burstein et al. defined four TNBC subtypes with unique copy number variations (CNVs) and distinct clinical outcomes. The LAR subtype exhibits the involvement of ER and the PR, FOXA, XBP1, and GATA3 genes, suggesting the evidence of ER activation in “ER-negative” tumors that might be related to 1% of cancer cells showing low levels of ER protein. LAR tumors may respond to traditional anti-estrogen therapies as well as to anti-androgens. The Mesenchymal (MES) subtype is characterized by deregulation of cell cycle, mismatch repair, DNA damage, and hereditary BC signaling pathways, such as overexpression of genes restricted to osteocytes (OGN) and adipocytes (ADIPOQ and PLIN1) and essential growth factors (IGF-1). Basal-Like Immune-Suppressed (BLIS) reveals downregulation of B, T, and NK cell immune-regulating and cytokine networks, showing the worst prognosis. Conversely, Basal-Like Immune-Activated (BLIA) exhibits upregulation of the genetic factor controlling B, T, and NK cell functions; activation; and high expression of STAT pathways and shows the best prognosis [33].

Le Du et al. revised TNBC molecular classifications and defined five potential clinical groups. BLBC is the most frequent subtype (25–80% of cases), characterized by DNA-repair gene deficiency and/or growth factor pathway overexpression, showing the highest pathological response (pCR) to chemotherapeutic treatment and sharing this feature with the Lehman-BL-1 group. They include the Lehman-BL-2 group into the mesenchymal subtype because it is enriched in growth factor/receptor tyrosine kinase pathways. The mesenchymal-like subtype includes the mesenchymal, mesenchymal stem-like, and claudin low subtypes and is characterized by EMT and cancer stem cell (CSC) characters, correlated with chemotherapy resistance [34]. The immune response signature was correlated with enriched levels of immune cell infiltration and good clinical response [35]. Tumor-infiltrating lymphocytes predict neoadjuvant response to chemotherapy [36,37]. Luminal/apocrine is enriched in hormonal-regulated pathways: AR overexpression might supply a lack of ER expression in steroid-signaling [12,27]. This group could include the LAR, Luminal A-B, Burstein’s LAR, and molecular apocrine subtypes, showing high expression of luminal gene, lack of basal-cytokeratin markers, and low proliferation rate [12,27]. The AR positivity, identified by IHC in a minimum of 10% of cancer cells, is predictive of about 30% of TNBCs with favorable prognosis [38,39,40]. Finally, they proposed the HER2-enriched subgroup as a distinct clinical entity but hypothesized the possibility that Luminal A and HER2-enriched subtypes can be brought together in a short period of time and suggested evaluating HER2-targeted therapies for this group of patients.

In 2016, the TNBC subtypes were refined by using histopathological quantification, laser capture microdissection, and RNASeq. The IM and MSL subtypes were excluded as TNBC subtypes as their features were associated to tumor-associated stromal cells and infiltrating lymphocytes and not to tumor cells [41]. Different clinical features and progression patterns have been demonstrated among the four TNBC subtypes. BL1 displays higher grade, lower stage, and amplified relapse-free overall survival (OS) of patient. LAR shows wider regional diffusion and preferentially distant metastasis to bone, whereas M tumors show that to lung. TNBC subtypes diverge in response to conventional neoadjuvant chemotherapy. The BL1 subtype displays the highest likelihood of getting a pCR, while LAR subtypes showed improved outcome against a reduced answer to neoadjuvant chemotherapy that could be caused by reduction in proliferation and luminal state. The clinical utility to stratify TNBC patients is that it could lead to selection of patients more responsive to chemotherapy [41].

Prado-Vazquez et al. analyzed clinical and expression profile data of TNBC through hierarchical clustering and probabilistic graphical models (PGM) showing two TNBC molecular classifications based on cellular type and immune activity. Based on the PGM, they distinguished four subgroups: CLDN-low, CLDN-high, basal-like, and LAR, consistent with the CSC hypothesis [42]. These classes define the differentiation process where the stem cell becomes carcinogenic and underline that CLDN-low is the less differentiated cancer, with respect to LAR, which is the most differentiated one. Strong molecular differences for therapeutic utility were identified. CLDN-low showed low alpha-amylase activity and regulation of actin cytoskeleton, and high haptoglobin activity. CLDN-high had low actin binding and high chemokine activity. Basal tumors had high cell adhesion and regulation of the actin cytoskeleton activity. LAR tumors showed low intensity regarding cell adhesion, G1/S transition of mitotic cell cycle, and chemokine action. Two additional TNBC subgroups have been defined, immune-positive and immune-negative, and demonstrated that the immune activation was significantly associated with good prognosis and positively influences the prognosis in the cellular LAR and CLDN-high groups [43].

Figure 1 illustrates the high molecular heterogeneity of TNBC as described by the cited authors. It shows the close relationship between TNBC and the basal-like phenotype, which is attested in each TNBC classification study, and the possibility to identify TNBC with other intrinsic phenotypes.

Molecular differences identified in TNBCs might be valuable for biomarker identification, drug discovery, and clinical trial design and could support the variations in histological types, prognosis, and patient’s outcome characterizing each TNBC molecular subtype. Notwithstanding some divergences, the TNBC molecular classifications previously discussed provide adequate evidence that there are four major subtypes demanding subtype-specific biological-based therapies. Predicted “driver” signaling pathways were pharmacologically targeted in clinical trials to prove if gene expression signatures can impact therapy selection (Figure 2).

Several studies have been performed to analyze clinical differences among TNBC molecular subtypes proving that BLP is of particular clinical interest, being diagnosed in younger women and showing higher grade, advanced clinical disease, and higher stage relative to non-basal TNBC. Although there are no differences in regional spread to lymph nodes and brain/lung metastasis between basal and non-basal TNBC [41], nonetheless, the M subtype displays a significantly higher frequency of lung metastasis vs. all other subtypes; significant incidence of bone metastasis has been identified in the LAR subtype, in accordance with the preference of hormonal-regulated cancers to metastasize to bone [67]. Basal-like TNBCs are largely classical ductal carcinomas, in contrast with lobular carcinomas, which are nearly exclusive to the LAR subtype, suggesting a potential role for AR signaling in lobular BC. Medullary histological types are common in all TNBC subtypes except for the M subtype, consistent with the lack of lymphocytic infiltration in these tumors. Metaplastic carcinomas are typical of basal-like and M subtypes. Specifically, the IM subtype displays the highest number of lymphocytes and lower involvement of lymph nodes, showing consequently the best OS and release-free survival (RFS). The LAR subtype is diagnosed in women of older ages compared to all other TNBC subtypes, showing lower grade and significant enrichment of lymph node metastasis [41]. Rakha et al. showed shorten BC-specific survival and shorter disease-free survival (DFS) in patients affected by TNBC with basal-markers matched to those with TNBC without basal-markers [68].

## 3. MicroRNA Dysregulation on TNBC with Basal-Like Phenotype: An Overview

Transcriptomic analysis provided the complex mRNAs and noncoding RNA expression patterns, which enables interpretation of the cell genomic codex. Applied to TNBC, it identifies deregulated miRNA expression patterns playing different roles as anti-oncomiR or oncomiR in tumor development, defining miRNA signatures related to prognosis [69,70] and stratification of TNBC into subclasses [71].

de Rinaldis et al. found 14 miRNAs related to TNBC prognosis influencing cell motility mechanisms. Moreover, 46 miRNAs were identified analyzing TNBC subtypes (according to Lehmann); 13 of them were overexpressed in BLBC, and miR-193a-3p had a prognostic role in TNBC [72]. Gasparini et al. analyzed a TNBC cohort divided by IHC into Core-Basal (CB) with EGFR and CK5/6-positive and into 5NP with five negatives markers, identifying a four-miRNA signature significantly dysregulated: miR-155, miR-493, miR-30e, and miR27a. This data allows stratification in high-risk and low-risk IHC subgroups, with CB patients having the worst prognoses. Both IHC and miRNA signatures showed prognostic significance in predicting patient outcomes based on different chemotherapy regimens [25,73]. TNBC with BLP was analyzed for identifying miRNAs associated to its biology and molecular features and related to putative therapeutic options. Recently, human miRNAome analysis was conducted in basal-like vs. non-basal-like TNBC, classified as quintuple negative BC by immunohistochemical basal-markers EGFR+ and CK5/6+. MiR-135b overexpression was highly related to TNBC with BLP and played a role in the TGF-beta, WNT, and HER2 pathways, confirming its involvement in TNBC with BLP pathogenesis. The miR-135b overexpression had a negative prognostic role that might be linked to a positive correlation with elevated proliferative index: modification of miR-135b expression could represent a therapeutic target in basal-like TNBCs [74]. Furthermore, previous studies showed miR-135b overexpression in TNBC and its downregulation in luminal tumors [18,75,76,77]. Kalecky et al., applying the Lehmann classifier to TCGA-derived TNBCs, found that BL1 was distinguished from BL2 through miR-17-92 cluster overexpression and consequently downregulation of several known miR-17-92 targets including INPP4B, thus supporting that TNBC could be distinct in subtypes by miRNA profiles which are linked to cancer-associated mRNAs [78].

Recently, researchers focused on model-based integrated analysis of miRNA/mRNA/lncRNA expression profiles on TNBC to identify subtype-specific miRNA signatures involved in oncogenic pathways and their potential role in survival outcome [79,80,81]. They revealed mechanisms of TNBC progression and identified novel prognostic and therapeutic biomarkers. A four-biomarker signature (miR-221, miR-1305, miR-4708, and RMDN2 mRNA) allowed the distinction into high- or low-risk TNBC groups and accounted for an independent prognostic survival factor. These four biomarkers are associated with cell cycle control and growth pathways, suggesting that their downregulation may be leading to aggressive and faster-growing tumors in patients classified with a poor prognosis [82].

Finally, miRNA signatures in exosomes were identified between HER2-positive and TNBC, showing the levels of exosomal miR-335, miR-376c, miR-382, and miR-433 overexpressed in TNBC compared to healthy women. These miRNA subtype-specific distributions in exosomes may be a selective exosomal packaging process. Exosome miRNA patterns were also associated with different clinicopathological parameters within the subgroups, and the levels of exosomal miR-374 were associated with a higher tumor size in TNBC [83].

### 3.1. Main miRNAs and Biology of TNBC with Basal-Like Phenotype

MiR-29c is a part of the miRNA family, including mir-29a and mir-29b-1/2, involved in neoplastic and nonneoplastic diseases [84,85,86,87,88]. MiR-29c is a fundamental regulator of many pathways, and its function in cancer is certainly challenging. MiR-29c downregulation distinguishes BLBC from other BC subtypes and divides them into two BLBC subsets. One of them shows reduced levels of multiple miRNAs’ regulatory and aberrant DNA hypermethylation. The genomic mechanism leading the DNMT3-b mediated aberrant-DNA hypermethylation in BC causes a posttranscriptional regulation defeat of DNMT3b by regulatory miRNAs [87]. Poli et al. identified the miR-29b-2/miR-29c promoter and detected hypermethylation of its CpG isles in BLBC compared to luminal cell lines. They demonstrated a significant inverse correlation between expression and methylation. Epigenetic control of the miR-29c promoter may offer a promising therapeutic target to address the invasive behavior of cancer [89]. MiR-29c is progressively lost during TNBC tumorigenesis and acts as a cancer suppressor in preneoplastic phase [90]. MiR-29c exerts growth inhibitory effects through direct binding and functional regulation of TGIF, CREB5, and AKT3. In BLBC patients with miRNA-29c downregulation, the mean survival was 60 months versus 95 months in patients with overexpression. Conversely, ectopic expression of miRNA-29c inhibits proliferation in preneoplastic cell models. Prevention strategies in miRNA-29c suppression have a greater biological role in the early preneoplastic stages of tumorigenesis [90].

Milioli et al. characterized two BLBC subgroups with distinct disease outcomes associated with differential expression and variability in miRNA and DNA copy numbers. MiR-29c-a showed greater expression in the Basal-I group associated with a better prognosis than Basal-II showing a poor prognosis. MiR-29c were associated to HJURP expression levels in G1 tumor [91]. MiR-29b-1-5p is downregulated in human BRCA1-associated BC, and BRCA1 impacts miR-29b-1-5p expression in BC cell lines, probably using promoter binding and transcriptional regulation. MiR-29b-1-5p, miR-664, miR-16-2, and miR-744 represent significant biomarkers for stratification of OS versus generally utilized clinical biomarkers such as lymph-node status and age at diagnosis in TNBC patients [92].

Epithelial cells undertaking EMT lose cell–cell relations and other epithelial traits compared to getting the migratory and aggressive phenotype during tumor diffusion and metastasis. EMT is related to cancer stem cell that could provide self-renewal and could initiate distant metastasis colonization [93]. The miR-200 family (miR-200f) is the main EMT regulator by using targeted silencing of transcriptional-EMT ZEB1 and ZEB2 inductors, which interact transcriptionally with miR-200f in a double negative feedback loop [94]. Specifically, metaplastic TNBC expresses lower levels of miR-200f than ER-positive tumors. MiR-200f overexpression is accompanied by upregulation of transcriptional EMT inducers and hypermethylation of the miR-200c-141 locus both in vitro and in vivo. Expression and methylation of miR-200f might represent a promising biomarker to establish the presence of EMT in BC [95]. miR-141/200c cluster overexpression in MDA-MB-231 cells increases VEGF-A secretion, which improves the migratory capacity through activation of the FAK and PI3K/AKT pathways [96]. The genes of the miR-200f grouped into two clusters miR-200b/200a/429 and miR-141/200c are prone to epigenetic regulation in cancer and normal tissue: a dynamic process that influences the miR-200f expression to regulate EMT and vice versa [97]. Several studies have correlated miR-200f and their mRNA targets with CSC properties. The reduction of expression levels of miRNA-200c-141, miR-200b-200a-429, and miR-183-96-182 is consistent in BC-CSC [98]. Downregulation of miR-200 in BC could induce conversion of breast epithelial cells into a stem cell phenotype [99].

MiR-34a expresses itself on luminal engagement and differentiation by inhibiting expansion of the mammary stem cell pool and early-progenitor cells, independently of p53. MiR-34a controls networks related to epithelial cell plasticity and luminal-basal conversion. MiR-34a chronic expression in triple-negative mesenchymal cells (enriched in CSC) could endorse a luminal-like variation program, limit the CSC pool, and inhibit the tumor. MiR-34a activation programs could provide a therapeutic opportunity for the BC subgroup enriched in CSCs and could poorly respond to conventional therapies [100].

Also, miR-205 is a negative regulator of EMT, targeting ZEB1 and ZEB2 mRNAs; its expression is lost in mesenchymal BC cell lines [101]. The expression of miR-205 is limited to the myoepithelial/basal cell compartment of normal mammary ducts/lobules and reduced or absent in the matching tumor [102], i.e., its downregulation is observed in TNBCs compared to normal tissue [103]. MiR-205 is downregulated in the highly aggressive TNBC subtype, while miR-205 overexpression strongly inhibits proliferation, cell cycle progression, and clonogenic potential in vitro and tumor expansion in vivo. A suppression activity is partially exerted through targeting E2F1 master regulator of cell cycle progression, the LAMC1 ECM component involved in cell adhesion, proliferation, and migration. P53 positively modulates miR-205 expression in different cells through direct binding on regulatory sequences upstream of the miRNA gene. A high frequency of P53 mutations in BLBC correlates with miR-205-5p downregulation [104,105]. MiR-205 is downregulated in highly migratory and invasive TNBC cells, and its re-expression displays a strong inhibitory effect on migration/invasion, CSC-like property, tumor growth, and metastasis. Mechanistic studies revealed that miR-205 effects were obtained through targeting ITGA5 and inhibiting the SRC/VAV2/RAC1 pathway. MiR-205-5p downregulation is associated with enhanced metastatic capability and worsening of OS [106]. Recently, miR-224-5p, miR-375, and miR-205-5p have been identified to distinguish non-basal, basal, and weak-basal TNBC subtypes, respectively, although they do not have any prognostic implication [107].

The Mir-17-92 cluster, also known as oncomiR-1, is the primary oncogenic miRNA found overexpressed in hematopoietic and solid tumors, including BC, involved as key regulators of proliferation and apoptosis cellular processes, and angiogenesis [108,109]. MiR-17-92 and its paralogical miR-106b-25 cluster are the most overexpressed miRNAs in BLBC. Their specific expression is associated with CNV, indicating that tumor genetic aberrations are responsible for a portion of miRNA expression differences in BC subtypes. Consequently, specific miRNA upregulation dictates transcriptional phenotypes as well as activation of oncogenic pathways in BLBC. Specific associations between miR-17-92/miR-106b-25 clusters and a large number of cancer-related pathways have been identified, including the MYC, mTOR, TGF-β, PTEN, and AKT pathways, and the EMT transcriptional signature by miR-17, miR-19a/b, and miR-106b. MiR-19b is related to components of focal adhesion and endothelium, while miR-92a is associated to the regulation of cytoskeleton [72]. BL1 and BL2 TNBCs [27] were identified using overexpression of miR-17-92 cluster and suppression of miR-17-92 targets involving INPP4B. In a BC cell line representative of BL1, a high level of miR-17/19a was identified, together with CDKN1A, IL1R1, FAM214A, and INPP4B downregulation, regulating cell cycle, apoptosis, and signal transduction [78]. Fazari et al. identified the involvement in TN BLBC of high miR-17, miR-19a, and miR-25 regulatory activity belonging to the oncogenic mir-17-92 and miR-200b cluster and their association with EMT process and distant metastasis-free survival. MiR-17-19a was related to a leukocyte trans-endothelial migration pathway, with similarities to metastasis and negatively correlated with CXCL12 [110]. MiR-19b-1-5p-17-3p and miR-200c-5p overexpressed in basal-II tumors relative to basal-I and controls and miR-19b-1-5p is associated with CXCR6 in low-grade BLBC, and miR-17 is associated with CTSK in high-grade BLBC [91].

An interesting archetype for a KLK5-miRNA-ECM network in BC [111] proposed that KLK5 directly leads to ECM and cell adhesion through its proteolytic activity and through miRNA-mediated pathways. KLK5 overexpression produces a miR-183-5p, miR-206, miR-181-c, and miR-19a overexpression and miR-935, miR-519a-5p, and miR-23b-5p downregulation. MiRNA underexpression could be regulated by specific enzymes involved in miRNA biogenesis which can be regulated in turn through several miRNAs upregulation targeting these enzymes. The merge of these networks have an amplified downstream effect on ECM molecules regulation, BC phenotype, and metastatic potential [111].

Eskandari et al. described the characteristics of a gene regulation network based on transcription factors, miRNAs, and their target genes, finding altered key regulators involved in BC subtypes and reducing OSs in each BC subtype. For the BL subtype, nine key regulators were identified including HMGA1, WT1, and FOXM1 adjusted at the top and STAT3 and TP63 at the bottom; mir-19a-3p, mir-106b-5p, mir-20a-5p, and mir-17-5p as key miRNAs were up-adjusted [112].

MiR-18a/b overexpression is strongly associated with BLBC characteristics, and patients with high proliferation, TNP, CK5/6-positive, tumor size >2 cm, and grade 3 cluster significantly together, showing miR-18a/b, miR-505, miR-25, and miR-106b overexpression and let-7b downregulation [113]. Neoplastic cells utilize autophagy to maintain intracellular homeostasis and to promote survival under stressful environments; moreover, this is an important mechanism of chemoresistance in TNBC cells. The paclitaxel (PTX)-resistant TNBC cells have a higher level of basal autophagy and miR-18a overexpression than the parent cells. Enforced miR-18a overexpression directly leads to increased autophagy levels, similar to the rapamycin effect (mTOR signaling inhibitor). MiR-18a overexpression decreases the expression of p-mTOR and p-p70S6, suggesting that miR-18a increases the autophagy level in TNBC cells via inhibiting the mTOR pathway, which is a mechanism contributing to PTX resistance [114].

MiR-221/222 is a cluster located on the X chromosome, where genomic abnormalities related to the BLBC pathogenesis often occurs. MiR-221/222 are at a 1 kb genomic distance with the same orientation and probably co-regulated [115]. This gene group is associated with EMT promotion, downregulation of ER, and tamoxifen resistance in BC [116]. MiR-221/222 have been recognized as BL-subtype-specific TNBC miRNAs, of which the overexpression reduced levels of epithelial-specific genes and improved mesenchymal-specific genes, favoring cell migration/invasion in an EMT characteristic manner. The transcriptional factor FOSL1, found in BLBC but not in luminal subtypes, expanded the transcription of miR-221/222, while their downregulation has been induced by EGFR or MEK inhibitors, placing miR-221/222 downstream of the RAS pathway. This model revealed that miR-221/222 overexpression is controlled by the EGFR/RAS/RAF/MEK/ERK2/FOSL1 axis and stimulates EMT by targeting TRPS1, which suppresses ZEB2, leading to E-cadherin downregulation. Because of the overexpression of miR-221/222, ZEB2 increases, permitting E-cadherin repression and vimentin upregulation promoting EMT [117].

MiR-200 family members are negatively regulated in aggressive BC tumors: a combined miR-221/222 and miR-200 signature with further EMT markers such as vimentin and E-cadherin can predict a model to identify patients with poor prognosis [101]. Specific miRNAs can promote transformation to more aggressive cancer phenotypes, promoting the clinically aggressive metastatic BC and suggesting that MEK-inhibitor and chemotherapy might provide a feasible strategy in BLBC patients [117]. Depletion of ADIPOR1, that is a miR-221 target, induced EMT in MCF10A cells, activated NFκB and JAK2/STAT3 pathways, and improved cell migration/invasion [118]. MiR-221 knockdown stopped cell cycle progression and encouraged apoptosis and proliferation in vitro and tumor growth in vivo. Silencing of miR-221 induces E-cadherin overexpression and SLUG and SNAIL downregulation in MDA-MB-231, BT-20, and MDA-MB-468 [119]. Shan et al. confirmed that miR-221/222 promotes the aggressiveness of BLBC by operating downstream of the RAS network and by triggering the EMT [120]. MiR-221/222 have also been shown to act as oncogenic by suppressing p27/Kip1 and p57 and by facilitating cell proliferation, self-renewal, and tamoxifen-resistance [121]. Table 1 summarizes all miRNAs involved in the biology of TNBC with basal-like phenotype.

### 3.2. miRNAs and BRCA Mutations

BRCA1-2 are onco-suppressor genes involved in double-stranded DNA break repair via the homologous recombination pathway (HRP) [122]. Germline BRCA mutations appear only in 3–5% of unselected BC patients [123], while TNBC with BLP are closely related to BRCA1 mutations, and 80–90% of BRCA1-abnormalities expressing cancers exhibited these phenotypes [124].

BRCA1 is a target of about 100 miRNAs and directly repress miRNA activity, i.e., miR-155, an oncomiR involved in BC formation and metastasis [125]. BRCA1 represses miR-155 expression by deacetylating histone H2A and H3 on the miR-155 promoter via HDAC2 with consequent dysregulation of cytokine pathways and promoting cellular transformation [126]. MiR-146a/miR-146-5p silence the BRCA1 which could trigger TN and BLBC development. A negative feedback has been described between miR-146a and BRCA1 mRNA, where miR-146a inhibits BRCA1 translation and, reversely, BRCA1 upregulates miR-146a [127]. In BRCA1-deficient cells, it was demonstrated that miR-146a levels were increased [128]. MiR-146a, miR-17, and miR-369 target BRCA1/2 mRNA and cause and influence BC pathogenesis [129]. MiR-21 is an oncomiR involved in BC progression: more than 70% of BCs amplify the miR-21 locus and show negative correlation between BRCA2 mRNA and miR-21 expression levels [130]. MiR-342 adjusts the BRCA1 expression by reducing the ID4 transcription factor involved in BC progression [131]; its reconstitution in TNBC cell lines led to caspase-dependent apoptosis but only in the presence of BRCA1 mutation. MiR-342 overexpression and mutant BRCA1 led to a synthetic lethal phenotype [132]. The interaction of miRNAs and proteins involved in HRP produces synthetic lethality that might be important for therapy success and BC pathogenesis. Moskwa et al. proposed that miR-182 downregulates BRCA1, and miR-182 manipulation expression in BC lines regulates susceptibility to poly-ADP ribose polymerase-1 (PARP) inhibition [133]. Finally, investigating the miRNA classifiers to identify BRCA germline mutation status, six miRNAs were assigned (miR-142-3p, miR-505, miR-1248, miR-181a-2, miR-25, and miR-340) [134].

### 3.3. miRNAs and Metastatic Process in TNBC with Basal-Like Phenotype

Although the diagnostic and prognostic features of miRNAs is now confirmed, deregulation of miRNAs in tumor progression and the related biological functions have not been fully defined.

Krutilina et al. reported that miR-18a, coded by the polycistronic gene MIR17HG, limits BLBC growth and lung metastases regulating HIF1A expression and hypoxic response. Orthotopic xenograft models of the metastatic variant MDA-MB-231 showed that downregulated miR-18a in BC cells spontaneously leads to lung metastases. The ectopic expression of miR-18a attenuates primary tumor expansion and decreases spontaneous lung metastases in contrast to miR-18a inhibition through a HIF1A-dependent pathway [173,174]. HIF1A is a miR-18a target; its expression influences hypoxic gene expression, cellular invasiveness, and sensibility to anoikis and hypoxia in vitro. An inverse correlation has also been identified between miR-18a and HIF1A mRNA expression levels in human BLBC compared to other BC subtypes [173].

Oncogenic EMT plays an important role in the cell’s ability to metastasize [175,176,177]. The miR-200f has emerged as a powerful regulator of EMT, is overexpressed in breast epithelial cells and luminal type carcinomas, and is lost in the more aggressive BLBC or TNBC [160]. Restoring miR-200c, aggressive TNBC improves anoikis sensitivity by identifying TRKB as a primary miR-200c target. An untargetable TRKB reverses the possibility of miR-200c to sensitize TNBC cells to anoikis [143]. TNBC cells in suspension upregulate TRKB and NTF3 to induce anoikis resistance. NF-kB guides the TRKB and NTF3 transcripts, and miR-200c defeats anoikis resistance by affecting this aberrant autocrine signaling loop [178]. NTF3 promotes TRKB [179] and represents a miR-200c target. Gene methylation regulates the miR-200c/ZEB1 axis, and chromatin remodeling of the miR-200c/miR-141 locus is influenced by ZEB1 contributing to cellular plasticity. In TNBC, methylation of the locus miR-200c/miR-141 is related to miR-200c downregulation and lymph node metastases and poor prognosis in TNBC. ZEB1 transcription factor overexpression suggests that the miR-200c/ZEB1 axis could represent a therapeutic focus in metastatic TNBC [180]. The ectopic expression of miR-200b suppresses TNBC invasion and metastases in a mouse xenograft BC model by inhibiting the Cα-protein kinase [181]. MiR-200a modulates TNBC migration using the EPHA2 oncogene as a regulator [182], while miR-200b-3p and miR-429-5p high levels negatively influence proliferation and migration/invasion of TNBC cells by inhibiting the LIMK1/CFL1 pathway [183], suggesting new perspectives for specific therapies in TNBC.

Exosomes-secreted miRNAs are essential in metastatic spread; BC cells can transmigrate through blood vessels, modifying the endothelial barrier. MiR-939 has been found overexpressed in BLBC and TNBC, influencing the lymph node status in predicting DFS probability. In vitro analysis has shown that miR-939 targets directly VE-cadherin, which leads to increased permeability of the Human Umbilical Vein Endothelial Cells (HUVECs) monolayer. MDA-MB-231 cells transfected with a miR-939 mimic released miR-939 into exosomes which, once internalized into endothelial cells, fostered the trans-endothelial migration of MDA-MB-231-GFP cells by breaking the endothelial barrier [170].

TGFβ1 is a regulator of HGF-induced and MET-mediated cell migration by C-ets-1-positive regulation and miR128-3p-negative regulation in BLBC cell lines and TNBC tissue [184].

MiR-31 is a metastatic suppressor that targets pro-metastatic genes (RHOA and WAVE3). Promoter hypermethylation of miR-31 has been identified as one of the main mechanisms of miR-31 silencing in BC and in basal subtype TNBC cell lines [185].

Downregulation of miR-424-5p is related to advanced clinical stage, increased tumor size, metastatic lymph nodes, distant metastases, and poor histological grade in BLBC patients. MiR-424-5p is an onco-suppressor that regulates proliferation and migration/invasion of cells through binding to the DCLK1 target and is associated with the aggressive state in BLBC [151].

### 3.4. miRNAs and Therapy in TNBC with Basal-Like Phenotype

TNBCs lack specific therapeutic targets; thus, chemotherapy remains the only therapeutic option [186]. The combination of miRNA-based therapeutic strategies with conventional therapies (chemotherapy, endocrine, or targeted therapy) could synergize anticancer effects, reducing toxicity and improving patient’s outcome.

Neijenhuis et al. identified miR-107 and miR-222 as regulators of the response to DNA damage and, by suppressing RAD51 expression, sensitize TNBC cells to PARP inhibitors [187].

EGFR overexpression characterizes 80% of TNBC and is associated with poor prognosis and decreased DFS [188]. MiR-141 targets the EGFR 3’-untranslated region to inhibit its translation, and the miR-141 promoter is repressed by KLF8. MiR-141 overexpression inhibits cell invasiveness, proliferation, and lung metastases in nude mice [189]. BRCA1 binds to the miR-146a promoter and promotes transcription, which consequently reduces EGFR expression [159].

VEGFR upregulation is associated with a decrease in DFS and OS in TNBC patients [188]. MiR-206 downregulation promotes invasion and angiogenesis in TNBC. MiR-141/200c overexpression promotes migration and invasiveness of TNBC cells through activation the FAK and PI3K/AKT pathways by secrete VEGF-A [96].

AR upregulation in ER-negative tumors is associated with a lower Nottingham grade, apocrine differentiation, and increased sensitivity to AR antagonists. Patients with AR-dependent TNBC decrease DFS, but the prognosis is better than those who are not AR-dependent. The mechanisms of chemotherapy in the treatment of LAR-TNBC were revealed in cancer cell lines exposed to 5-fluorouracil plus ixabepilone, miR-122a, miR-145, and miR-205 overexpression and miR-296 downregulation as well as other alterations of miR-221, miR-210, miR-21, and miR-10b [190].

Recent studies show that lncRNAs-LINP1, overexpressed in TNBC, improves double-stranded DNA repair by coordinating the NHEJ pathway, and that LINP1 block, regulated by p53 and EGFR, increases the radiotherapy sensitivity of TNBC cell [191,192]. Lapatinib plus imatinib therapy in TNBC leads to the synergistic repression of upregulated lncRNA-HOTAIR, which regulates the proliferation and invasion of cells [193]. Estradiol promotes HOTAIR through its GPER receptor by inhibiting miR-148a, and estrogen-induced migration of tumor cells is reversed by eliminating HOTAIR in TNBC cells [194].

Repeated chemotherapy treatments induce resistance of TNBC cells leading to recurrence and metastasis [195], and the dysregulation of miRNAs is linked to resistance development [196,197,198]. MiR-105 and miR-93-3p may interact with secreted frizzled-related protein 1 to activate WNT signaling, which causes chemoresistance in TNBC cells [199]. MiR-873 promotes gemcitabine resistance through ZEB1 overexpression, becoming a promising predictor for gemcitabine sensitivity in TNBC patients [200].

Data on the miRNA expression profile and treatment response or drug resistance in the TNBCs with BLP is scarce. MiR-125a-3p may induce changes in the HER2 pathway in the BLBC, thus sensitizing cells to anti-HER2 therapies; its overexpression reduces the migratory capacity of MDA-MB-231 cells and leads to high levels of HER2 transcripts. The induced HER2 responses to trastuzumab underwent internalization and intra-lysosomal degradation, and its use negatively impacts migratory capacity and induced apoptosis. Mouse model demonstrated a synergistic consequence for miR-125a-3p and trastuzumab: miR-125a-3p-induced tumors treated with trastuzumab were significantly smaller than controls. MiR-125a-3p, alone or in combination with trastuzumab, might possibly represent a new treatment for BLBC [201].

The impact that neoadjuvant platinum-based chemotherapy (NCT) has on miRNA expression profiles in basal-like TNBC patients shows a pre-NCT “cancer model” that returns to a “healthy control model” after NCT: with miR-34a-5p overexpression after NCT, while pre-NCT and controls had similar expression, miR-664b-3p was downregulated after NCT, miR-144-3p, miR-144-5p, miR-126-5p, and hsa-let-7d-5p were overexpressed before NCT, while post-NCT signatures matched the controls. NCT has a significant influence on miRNA expression that appears to be a reasonable candidate for minimally invasive basal-like TNBC detection and therapy monitoring. The results suggest that changes in miRNA expression and blood miRNA profiles monitored over time have the potential to predict pCR [202].

Mifepristone (MIF) suppresses TNBC tumor growth and xenograft lines in NOD-SCID mice. MIF reduces the TNBC-CSC population through downregulation of KLF5, an overexpressed stem-cell transcription factor in basal-like TNBC. MIF suppresses KLF5 by inducing miR-153 expression. Accordingly, miR-153 reduces CSC and the miR-153 inhibitor rescued MIF-induced downregulation of the KLF5 protein level and CSC ratio. These prove that MIF inhibits TNBC via the miR-153/KLF5 axis and represent a treatment for TNBC [203].

MiR-100 is downregulated in all BC subtypes, especially in Luminal A, which reacts to hormonal treatments but not to chemotherapy, including microtubules-targeted PTX. In the MDA-MB-231 basal-like cell line, the suppression of miR-100 compromises the influence of PTX on cell cycle arrest, multinucleation, and apoptosis. Patients with miR-100 downregulation have a worse OS, suggesting that miR-100 dictates BC PTX sensitivity [204]. In addition, Petrelli et al. identified an additional role for miR-100, demonstrating that miR-100 reduces the preservation and expansion of BC-CSCs in BLBC through downregulation of PLK1. MiR-100 promotes BC-CSC differentiation, moving BLP into the luminal subtype, inducing ER expression and increasing hormone-therapy sensitivity. MiR-100 downregulation is an unfavorable prognostic factor related to gene signatures of high-grade undifferentiated cancers. These results suggest possible new treatments, which might make BC aggressive cells reactive to standard procedures [205].

Strategies to reprogram aberrant miRNA networks in cancer would be more effective considering that proliferation, invasion-migration, and acquisition of CSC properties represent biological processes to improve cancer progression and poor clinical patient’s outcome. Between the TNBC molecular classes, more genomic-epigenomic variations have been identified for the basal-like subtypes; Figure 3 represents a miRNAs/mRNA complex network between pathways involved in growth; proliferation; apoptosis; and processes such as EMT, CSC properties, and suppression of ER mRNA levels involved in the development of this aggressive BC subtype. While mRNA drugs inhibitors are well-known, as described in the figure, very little is known about its mechanisms and drug development to control oncomiR/anti-oncomiR. Currently, therapeutic methods to silence/activate oncomiR/anti-oncomiR have been hypothesized and studied, such as epigenetic silencing of cognate host genes, development of antagomirs or mimic-miRNAs conveyed by nanoparticles, and identification of specific transcription factors for miRNAs which could be hypothesized to modulate miRNA expression variations in cancer.

## 4. Conclusions

This review described the complexity of the miRNA network in TNBC with BLP, the most aggressive BC variants, which has not been thoroughly characterized up to date. Despite the efforts made for a molecular taxonomy and finding of prognostic and predictive indicators on TNBC based on gene expression profile, a limited impact on clinical applications has been achieved. TNBC could greatly benefit from progress in this field, considering the absence of therapeutic goals and the need to develop personalized therapeutic approaches.

Consequently, a targeted approach on epigenetic mechanisms responsible for its aggressiveness and clarification of the molecular mechanisms of specific miRNAs in the TNBC subtype are warranted, hoping that miRNAs will become therapeutic targets that could enhance, in combination with chemotherapy agents, positive effects on TNBC subtypes. Currently, TNBC with BLP is the only subtype for which we might suggest some therapeutic hypotheses based on miRNA/mRNA integration.

## Figures and Tables

**Figure 1 cancers-12-03298-f001:**
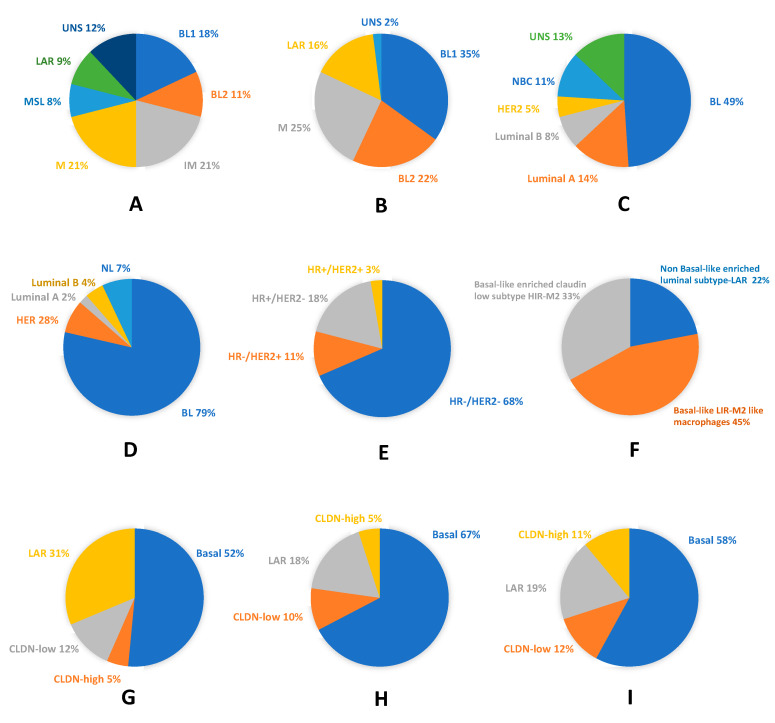
Molecular and phenotypic heterogeneity of Triple Negative breast cancer (TNBC): (**A**) pie chart showing frequencies and molecular TNBC type-6 classification (according to Lehmann et al. 2011) [13]; (**B**) pie chart displaying frequencies and molecular TNBC type-4 classification (according to Lehmann et al. 2011) [41]; (**C**) TNBC assigned to intrinsic molecular breast cancer (BC) subtypes (according to Lehmann et al. 2011) [13]; (**D**) pie chart showing molecular subclassification and frequencies of basal-like phenotype (BLP) in TNBC (according to Prat et al. 2013) [44]; (**E**) pie chart displaying molecular subclassification and frequencies of TNP in BLBC (according to Prat et al. 2013) [44]; (**F**) pie chart showing the fuzzy clustering of TNBC (according to Jezequel et al.) [45]; (**G**) pie chart displaying the probabilistic graphical model of TNBC (according to Prado-Vazquez et al.) [42]; (**H**) pie chart showing the overlapping between the probabilistic graphical model of TNBC and the immune activity negative group (according to Prado-Vazquez et al.) [42]; and (**I**) pie chart displaying the overlapping between the probabilistic graphical model of TNBC and the immune activity-positive group (according to Prado-Vazquez et al.) [42]. BL, basal-like; BL1, basal-like 1; BL2, basal-like 2; CLDN, Claudin; HER2, human epidermal growth factor receptor 2; HR−, hormone receptor negative; HR+, hormone receptor positive; IM, immunomodulatory; LAR, luminal AR; M, mesenchymal; MSL, mesenchymal stem-like; NBC, non-basal-like; NL, normal-like; and UNS, unstable.

**Figure 2 cancers-12-03298-f002:**
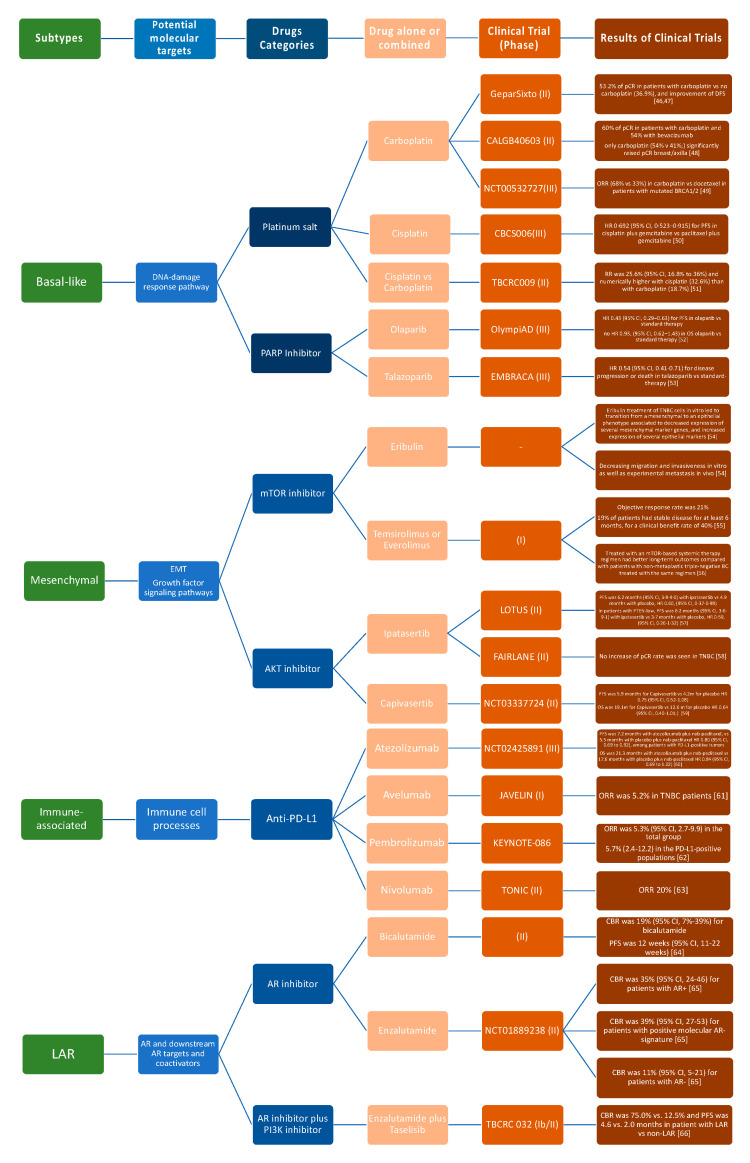
Current advances in systematic treatment for patients affected by different TNBC subtypes [46,47,48,49,50,51,52,53,54,55,56,57,58,59,60,61,62,63,64,65,66].

**Figure 3 cancers-12-03298-f003:**
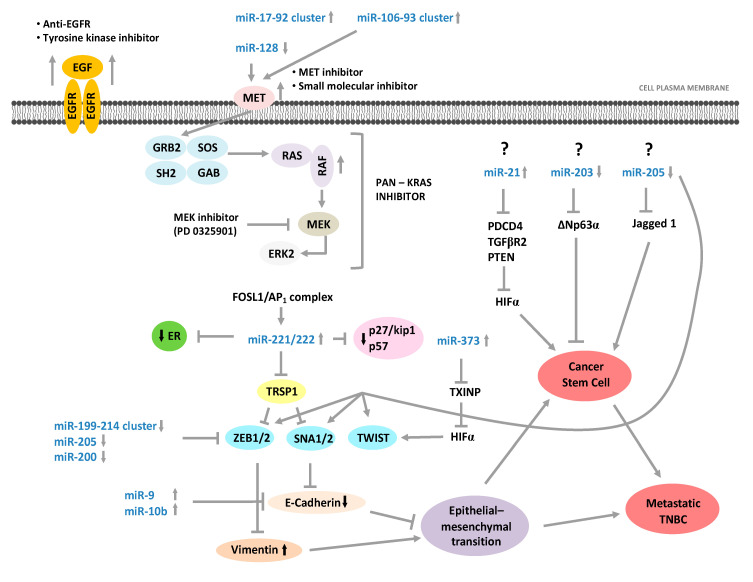
miRNAs/mRNA complex network in TNBC with basal-like phenotype and potential sites of therapeutic interventions.

**Table 1 cancers-12-03298-t001:** miRNA biological function and effect in TNBC with basal-like phenotype.

Function	MiRNAs	Validated Targets	Biological Function and Effect	Reference
**Tumor suppressive** **(inhibition)**	miR-34c	CCND1, CDK4, CDK6, CDC23	Cell proliferation and apoptosis	[135]
	miR-204-5p		Poor clinical outcome; metastasis and anoikis sensitivity	[136]
	miR-126-3p		Cell proliferation and invasion; TNBC outcome; overall survival	[18,137,138,139]
	miR-128-3p/5p	MET	Cell migration and invasion; relapse-free survival; discrimination among TNBC with basal vs. non-basal subtypes	[107,140]
	miR-143/miR-145	MUC1	Cell proliferation and invasion	[141]
	miR-29a, miR-29b, miR-29c, miR-26a, miR-26b, miR-148a, miR-148b	DNMT3b, TGIF2, CREB5, AKT3	DNA methylation; overall survival; cell proliferation and colony formation; modulation of ER/PR/HER2/CK5/6 expression	[72,87,89,90,92,113]
	miR-149	RAP1A, RAP1B, VAV2	Metastasis; higher tumor stage	[142]
	miR-155, miR-493, miR-30e, miR-27a		Discrimination among TNBC with core-basal or non-core-basal subtypes and among TNBC patients with high- and low-risk prognoses; diagnostic and prognostic indicators; prediction of outcomes of patient treatment with anthracycline or anthracycline plus taxanes	[25]
	MiRlet7c, miR-125b, miR-126, miR-127-3p, miR-143, miR-145, miR-199a-3p		Basal cell type-specific miRNAs	[18,139]
	miR-200 family (miR-200b, miR-200a, miR-429, miR-200c, miR-141)	ZEB1, ZEB2, SUZ12, SNAI1, SNAI2, CDH11, CFL2, SEC23A, EPHA2, TRKB	Modulation of EMT-transcriptional inducers; modulate luminal cell type-specific miRNAs; enhanced stem cell self-renewal; cell migration and invasion; suppression of anoikis resistance; poor survival	[95,139,143,144]
	miR-203a	ΔNp63α, SLUG, AXL	Cell growth and motility; discrimination among TNBC with basal vs. non-basal subtypes; cancer stem cell-like property	[145]
	miR-205–5p (miR-205)	KLF12, ITGA5, E2F1, LAMC1, ZEB1, ZEB2, Jagged1, PTPRM	Cell invasion, migration, and apoptosis; cancer stem cell-like property; cell growth and metastasis; EMT	[104,105,106,107,136,144]
	miR-30a-3p, miR-30a-5p, miR-199a-5p, miR-30c-5p	CDH1, YAP1, TWIST, β3-INTEGRIN, ZEB1, ZEB2	Overall survival and relapse-free survival; discrimination among TNBC with basal vs. non-basal subtypes; modulation of the expression pattern of EMT-related genes; cell proliferation, migration, invasion, and apoptosis; sustainment of angiogenesis	[18,107,146,147]
	miR-206	ER	Positive correlation with the expression level of ER and PR and negative correlation with HER2; negative correlation with TNM staging	[148]
	miR-20b	STAT3, HIF1A	Reduction in the expression of the vascular endothelial growth factor (VEGF)	[149]
	let-7a-b-c-5p	DICER	Cell proliferation and metastasis; modulation of ER/PR/HER2/CK5/6 expression; disease-free survival and overall survival	[18,72,113,146,150]
	miR-424-5p	DCLK1	Cell proliferation, migration, and invasion	[151]
	miR_375	SHOX2, LDHB, CPNE8, QKI, EIF5A2	Reverses the resistance to tamoxifen in breast cancer cells; EMT; cell proliferation, migration, and invasion; modulation of ER/PR/HER2/CK5/6	[107,113,152,153]
	miR-425			[72]
	mir-149, mir-218, mir-374b		Cell proliferation, invasion, and colonization; discrimination between BRCA1 and sporadic basal cancers; TNBC outcome; modification of the therapeutic effects of 5-fluorouracil and cyclophosphamide treatments	[75,138]
	miR-100, miR-99, miR-214, miR-342			[18]
**Oncogenic (promotion)**	miR-182	CFN, PFN, BRCA1	Cytoskeleton reorganization; cell proliferation and invasion	[154]
	mir-198		Discriminate between BRCA1 and sporadic basal cancers	[75]
	miR-183	ITGB1, COL12A1, COL21A1, DICER1, AGO1-2	Metastasis; discrimination of molecular subtypes of breast cancers; overall survival	[111]
	miR-206	DICER1	Cell growth	[111]
	miR-181	ITGB1, TGFβR3	Strong expression in p53 mutant BLBCs; metastasis and reversion of anoikis resistance through their negative regulation in the TGF-beta signaling pathway	[111,136,155]
	let-7d-3p, miR-324-5p, miR-203b	DNMT3b	Overall survival and relapse-free survival; cell migration and invasion	[87,107]
	miR-95-3p	SNX1	Decreased OS and RFS in patients treated with anthracycline-based chemotherapy; cell proliferation, migration, invasion, and apoptosis	[107]
	miR-21	TPM1, PDCD4, PTEN, TGFR2	Negative correlation with the expression levels of ER and PR and positive correlation with HER2; cell growth, invasion, and metastasis; positive correlation with TNM staging; EMT; cancer stem cell-like property; cytoskeleton reorganization	[148,154]
	miR-130a/b-3p	PTEN, PIEZO2	Mediate drug resistance; cell proliferation	[146,156]
	miR-17, miR−18a, miR−19a, miR−20a, miR−19b e miR-92amiR-18b	Luminal specific-gene sets, tumor low-grading gene sets, ER, PI3K, MET, DROSHA, IL1R1, NPP4B, CDKN1A¸ FAM214A, E2F1, PTEN, mTOR, p70S6, CXCL12, HOXA9, AQP5, RUNX3	Cell proliferation; modulation of ER/PR/HER2/CK5/6 expression; endocytosis; cell migration, adhesion, remodeling; distinguishment between BL1 and BL2 subgroups; increased autophagy involved in PTX-resistance; metastasis; expression in high-grade TNBC	[18,72,75,78,110,111,112,113,114,146,157,158]
	miR-211-5p			[146]
	miR-500a-3p			[146]
	miR-505		Cell proliferation; modulation of ER/PR/HER2/CK5/6 expression	[113]
	miR-106a-b, miR-106b-25 cluster (miR-106b, miR-93)	Luminal specific-gene sets, tumor low-grading gene sets, PI3K, MET	Cell proliferation; modulation of ER/PR/HER2/CK5/6; endocytosis; cell migration, adhesion, and remodeling; expression in high-grade TNBC	[18,72,75,110,112,113,158]
	miR-146a, miR-146b-5p	BRCA1-2, TNF, FADD, TRADD, IRAK1, NFKBIA	Cell proliferation and apoptosis homologous recombination miR-146a strongly expressed in p53 mutant BLBCs; BRCA1-deficiency direct decrease in miR-146, which consequently determines EGFR overexpression	[18,127,139,155,159]
	miR-27b-3p		cell proliferation and growth; TNBC outcome	[138]
	miR-221/mir-222	TRPS1, ER, p27/Kip1, p57, SOCS1, CDKN1B, DNMT3b, ADIPOR1	EMT; cell invasion and migration; progression of the more aggressive ER-negative basal phenotype; conference of resistance to tamoxifen; aberrant DNA hypermethylation	[87,120,149,160]
	miR-34b		Overall survival and relapse-free survival; cell proliferation, differentiation, and aggressiveness	[161]
	miR-362-5p	Sema3A	Cell proliferation, migration, and invasion	[162]
	miR-155	TSPAN5	Promotion of stem cell proliferation and cellular proliferation	[18,163]
	miR-150			[18]
	miR-142-5p miR-142-3p	PTEN	Cell proliferation and apoptosis	[18,75,164,165]
	miR-135b		Cell proliferation, AR status, and age	[18,74,75,76,139]
	miR-421	PIEZO2, PDCD4	Cell proliferation	[156]
	miR-454-3p	PIEZO2, AKT	Cell proliferation, migration, invasion, and apoptosis	[156]
	miR-301a-3p,	PTEN, ER, PIEZO2	Metastasis; suppression of estrogen signaling; poor prognosis	[156]
	miR-196a-5p,	PIEZO2, SPRED1	Cell growth and metastasis	[156]
	miR-455-5p	CDKN1B	Expression in exosomes; poor prognosis in the BLBC subtype; cell cycle process; cell proliferation, invasion, and migration	[166]
	miR-1255a	SMAD4	Expression in exosomes and original cells samples; poor overall survival	[166]
	miR-134		Expression in p53 mutant BLBCs	[155]
	miR-934	ER, FOXA1, GATA3 (genes involved in luminal lineage)	VGLL1 and miR-934 overexpression maintaining the luminal progenitor phenotype, at least in part mediated by their direct modulation of ER	[167]
	miR-10a-b	RB1CC1, CHN1	Metastasis; EMT; Contribution to the invasive progression of DCIS-associated myoepithelial cells into IDC via TGF signaling activation.	[111,168,169]
	miR-939	VE-CADHERIN	Disease-free survival; disrupting endothelial junctions and impairment of endothelial cell function; exosome-associated miR-939 increase in tumor cell trans-endothelial migration	[170]
	miR-9	CHN1, PDGFR	EMT; formation of vascular-like structures; disease-free survival and distant metastasis–free survival	[171,172]

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
