# Peer review of "Modulatory Role of microRNAs in Triple Negative Breast Cancer with Basal-Like Phenotype"

_cancers, 2020, doi:10.3390/cancers12113298_

Round 1

Reviewer 1 Report

Here the authors seek to improve the understanding  and therapeutic targeting of TNBC through the evaluation and review of miRNAs. The use of miRNA in cancer biology is a large area of investigation, and specifically increased understanding in the TNBC subtype would advance the field, the overall organization and structure diminished enthusiasm for this manuscript. The review reads as a long continual list of miRNAs in cancer, the relevance and significance to specific TNBC phenotypes is hard to gleam from this as it is written. Some restructure and reorganization would be helpful to the reader. Furthermore the addition of additional tables, visual aids of significant miRNA and miRNA targets in the TNBC subtype would enhances the enthusiasm of this manuscript.

  1. Authors close with the statement that they suggest miRNAs should be used for therapeutic targets, if this is the suggested case being made, authors should include information/chart on TNBC molecular subtypes and miRNA profiles associated with these subtypes (if known) along with target genes of interest that would suggest an miRNA could be exploited for therapy
  2. Chart/table for TNBC subtypes and characteristics with regard to response to therapy should be included
  3. Subject header 3 “MicroRNAs Dysregulation on Triple Negative Breast Cancer with Basal-Like Phenotype: An Overview”
    1. Initial discussion is on ER regulation but and endocrine resistance with little focus to how miRNA-ER regulation can regulate and progress to loss of receptor status. Because of this, the first four-five paragraphs seem irrelevant to the topic of TNBC.
    2. Furthermore in this section, some statements do not back the data, for example for ER regulation by miRNA authors use this example: “MiRNAs control ER expression and the hormonally-related signaling pathways in BC. Miller et 232 al. reported overexpression of miR-221/222 in HER2/neu-positive vs HER2/neu-negative” this is more an example of HER2 expression and there are much better examples of miRNA and ER expression
    3. There is no overall evaluation or discussion on if a miRNA TNBC signature exists, this would be an important starting point for this section and should be addressed. In addition the authors speak of TNBC heterogeneity but do not address if there are microRNA signatures for the TNBC subtypes.

Minor

  1. Authors should check for sentence structure and grammar, words are missing or in incorrect order. Examples:
    1. “cytokeratin such 50 as EGFR and CK5/6” reads as if suggesting EGFR a cytokeratin
    2. Other sentences missing words for complete sentences. Example in abstract “unresponsive and 27 unable to endocrine therapy” and “The BCs classification as luminal or basal-like subtype dependent on expression of different 116 cytokeratins”
    3. “In TNBC, MiR-107”
  2. Authors should check for abbreviation use. Examples:
    1. TN/ TNP with no abbreviation designation in abstract.
    2. Use of TNBC, BLBC in rest of manuscript without designation.
  3. Authors should check use of paragraph structure many points where one sentence is sued to comprise an entire paragraph. This seems unconventional.
  4. Authors should reference figure 1 in the text. As is, this figure is hard to understand and the significance of its finding are lost on the reader. Labels for what charts represent are needed for Figure 1, currently it is hard to interpret.
  5. Standardization for gene ID should be used, example: authors switch between ER, ER-alpha, ESR1
  6. Authors use * designations for miRNA which are not the current convention.

Author Response

We are grateful for your consideration of this manuscript, and we thank the reviewer who have permitted us to improve the quality of our work with their suggestions. All the comments we received on this study have been considered and we have addressed all of them in our point by point reply below.

  1. Authors close with the statement that they suggest miRNAs should be used for therapeutic targets, if this is the suggested case being made, authors should include information/chart on TNBC molecular subtypes and miRNA profiles associated with these subtypes (if known) along with target genes of interest that would suggest an miRNA could be exploited for therapy

We agree with Reviewer comments, the development of cancer therapies targeted epigenetic mechanisms is fascinating. Strategies to reprogram aberrant miRNA networks causing cancer would be more effective considering that proliferation, invasion-migration, acquisition of stem cell-like properties represent biological processes to improve cancer progression and poor clinical patient’s outcome. Nowadays, more is known about mRNA drivers in cancer, already used to treat different tumor types such as bevacizumab in breast cancer HER2-overexpressing. Conversely, no targeted therapies are available for TNBC.

Between the TNBC molecular classes, a lot of genomic-epigenomic variations has been identified in basal-like subtypes, taking into account your suggestion we have tried to represent a miRNAs/mRNA complex network using growth, proliferation and apoptosis pathways and processes such as epithelial-mesenchymal transition, acquisition of stem cell-like properties, suppression of ER mRNA levels involved in the development of aggressive breast cancer subtypes. While mRNA drugs inhibitors are known, very few is known about mechanisms and drugs development to control oncomiR/anti-oncomiR. Currently, therapeutic methods to silence/activate oncomiR/anti-oncomiR have been hypothesized and studied, such as epigenetic silencing of cognate host genes, development of antagomirs or mimic-miRNAs conveyed by nanoparticles, identifications of specific transcription factors for miRNAs could be hypothesized to modulate miRNAs expression variations in cancer. For all the above reasons, we have introduced the Figure 3 at the end of chapter 3.

  1. Chart/table for TNBC subtypes and characteristics with regard to response to therapy should be included

We are grateful to reviewer for this useful suggestion, considering the necessity of unambiguous biomarkers and therapeutic options for TNBC, which is the most clinical aggressive breast cancer variant, and its deep heterogeneity, a graph that reassumes the current therapeutic clinical trials and their results could be of interest for scientific community. For this reason, we have introduced Figure 2, which describes the more recent information about molecular targets, drug family, drugs, and response to therapy for each molecular TNBC subtypes, emphasizing the strongly therapeutic differences between TNBC subtypes.

  1. Subject header 3 “MicroRNAs Dysregulation on Triple Negative Breast Cancer with Basal-Like Phenotype: An Overview”
    1. Initial discussion is on ER regulation but and endocrine resistance with little focus to how miRNA-ER regulation can regulate and progress to loss of receptor status. Because of this, the first four-five paragraphs seem irrelevant to the topic of TNBC.
    2. Furthermore in this section, some statements do not back the data, for example for ER regulation by miRNA authors use this example: “MiRNAs control ER expression and the hormonally-related signaling pathways in BC. Miller et 232 al. reported overexpression of miR-221/222 in HER2/neu-positive vs HER2/neu-negative” this is more an example of HER2 expression and there are much better examples of miRNA and ER expression

We agree with these suggestions and removed the introduction of paragraph “MicroRNAs Dysregulation on Triple Negative Breast Cancer with Basal-Like Phenotype: An Overview”.

    1. There is no overall evaluation or discussion on if a miRNA TNBC signature exists, this would be an important starting point for this section and should be addressed. In addition, the authors speak of TNBC heterogeneity but do not address if there are microRNA signatures for the TNBC subtypes.
    2. To better reply to this comment, we revised the literature and strongly modified the paragraph “MicroRNAs Dysregulation on Triple Negative Breast Cancer with Basal-Like Phenotype: An Overview”. We discussed miRNA TNBC signatures which greatly support the genomic TNBC heterogeneity, although the published data prevalently concern TNBC with basal-like phenotype, considering its high frequency, high aggressiveness respect to other TNBC subtypes. MiRNA signatures in TNBC with basal-like phenotype underline its further heterogeneity, i.e. miRNA signatures are able to distinguish TNBC basal-like with good and worse prognosis. We introduced these modifications in “MicroRNAs Dysregulation on Triple Negative Breast Cancer with Basal-Like Phenotype: An Overview” pages. 7-8, lines 256-297.

  1. Authors should check for sentence structure and grammar, words are missing or in incorrect order. Examples:
    1. “cytokeratin such 50 as EGFR and CK5/6” reads as if suggesting EGFR a cytokeratin

We are grateful to the reviewer for the careful reading of our text: we have erroneously defined EGFR a cytokeratin. We have change the sentence as follows: “BLBC is characterized by gene expression usually found in basal or myoepithelial mammary cells, such as high molecular weight cytokeratin, prevalently CK5/6, but also by EGFR expression, and about 75% of them referred to TNP, being also ER/PR/HER2 negative by immunohistochemistry (IHC).” (in Introduction section pag. 2, lines 68-71)

    1. Other sentences missing words for complete sentences. Example in abstract “unresponsive and 27 unable to endocrine therapy” and “The BCs classification as luminal or basal-like subtype dependent on expression of different 116 cytokeratins”

We modified the two sentences as follows “The TN phenotype in basal-like breast cancer (BLBC) make it unresponsive and unable to endocrine therapy” (in Abstract section pag.1, lines 41-42)

“The BCs classification as luminal or basal-like subtype dependent on typical proteins expression. Luminal A-B BC express protein of luminal epithelial cells, called ‘‘luminal group”,  such as luminal cytokeratins (CK8, 18), ER, GATA3;  BLBCs express high weight basal cytokeratins (CK5/6, 14, 17) and EGFR, and/or c-KIT” (in Molecular Classifications of TNBC: Clinical Outcome Implications section pag. 3, lanes 125-128)

    1. “In TNBC, MiR-107”

We changed the phrases with “Neijenhuis et al. identified miR-107 and miR-222 as regulators of the response to DNA damage, and by suppressing RAD51 expression sensitize TNBC cells to PARP inhibitors” (in MicroRNAs Dysregulation on TNBC with Basal-Like Phenotype: An Overview section pag. 17, lanes 519-520)

We have checked all text for grammar, spelling, punctuation mistakes and missing words.

  1. Authors should check for abbreviation use. Examples:
    1. TN/ TNP with no abbreviation designation in abstract.

We have modified the text, as suggested.

    1. Use of TNBC, BLBC in rest of manuscript without designation.

We have modified the text, as suggested.

  1. Authors should check use of paragraph structure many points where one sentence is sued to comprise an entire paragraph. This seems unconventional.

We have edited the text and paragraph structure in accordance with the reviewer's suggestions.

  1. Authors should reference figure 1 in the text. As is, this figure is hard to understand and the significance of its finding are lost on the reader. Labels for what charts represent are needed for Figure 1, currently it is hard to interpret.

According with reviewer suggestion, we referenced the Figure 1 in the text “The Figure 1 represents the high molecular heterogeneity of TNBC as described by cited Authors. It shows the close relationship between TNBC and basal-like phenotype, which is attested in each TNBC classification study, but also the possibility to identify TNBC with other intrinsic phenotypes.” (in Molecular Classifications of TNBC: Clinical Outcome Implications section pag. 6, lanes 223-226)

  1. Standardization for gene ID should be used, example: authors switch between ER, ER-alpha, ESR1

We standardized the use of gene ID in all text.

  1. Authors use * designations for miRNA which are not the current convention.

We replaced the old miRNA nomenclatures with new designation using miRBASE:

miR-19b-1*  >  miR-19b-1-5p

miR-17*  >  miR-17-3p

miR-200c*  >  miR-200c-5p

Reviewer 2 Report

The manuscript by Angius and colleagues, offer a summary of the current knowledge of the epigenetic discoveries, related to miRNA expression profiling and their regulation in relationship to TNBC biological context with basal-like phenotype. In particular the manuscript reviews the emerging roles of miRNAs in pathogenesis and prognosis of basal-like TNBC and discuss their potential as biomarkers in clinical utility including their possible theoretical role as new targets for therapies development in this particularly aggressive breast cancer subtype.

This review is quite well organized and presents a quite comprehensive point of view of the literature, however, the manuscript presents some minor points that need to be addressed and clarified before it can be further considered for publication.

Minor points

- I would suggest to the authors to include more words concerning Triple Negative Breast Cancer molecular characteristics and heterogeneity, moving part of the paragraph 2 in to the introduction, as well as endocrine therapy treatments and resistance in the “Introduction section” to give at beginning a better view of the Triple Negative Breast Cancer context that can help also  readers not familiar with this field before analysing in deep the BL subtype.

- Concerning Table 1, I would suggest to the authors to modify the header of the table for a more clear and easy readability. I would include the term “miRNA function and miRNA Biological function and effect” since not for all miRNAs validated targets are reported.

-I would suggest to include a paragraph concerning the differences between TNBC with basal vs non-basal like phenotype to better highlight the BL specificities in relation to the reported data.

-There are some inaccuracies and typos within the text that need to be corrected.

Author Response

- I would suggest to the authors to include more words concerning Triple Negative Breast Cancer molecular characteristics and heterogeneity, moving part of the paragraph 2 in to the introduction, as well as endocrine therapy treatments and resistance in the “Introduction section” to give at beginning a better view of the Triple Negative Breast Cancer context that can help also readers not familiar with this field before analysing in deep the BL subtype.

We are grateful to the reviewer for this suggestion, we have deeply modified the Introduction section including additional data concerning TNBC characteristics, heterogeneity, therapeutic options, and resistance to classic-innovative treatment applied in breast cancer (in Introduction section pag. 2, lanes 58-81)

- Concerning Table 1, I would suggest to the authors to modify the header of the table for a more clear and easy readability. I would include the term “miRNA function and miRNA Biological function and effect” since not for all miRNAs validated targets are reported.

We modified the title of Table 1 as suggested “miRNA biological function and effect in TNBC with basal-like phenotype”.

We are grateful for your consideration of this manuscript, and we thank the reviewer who have permitted us to improve the quality of our work with their suggestions. All the comments we received on this study have been considered and we have addressed all of them in our point by point reply below.

-I would suggest to include a paragraph concerning the differences between TNBC with basal vs non-basal like phenotype to better highlight the BL specificities in relation to the reported data.

As suggested by the Reviewer, we have integrated the Molecular Classifications of TNBC: Clinical Outcome Implications section with a paragraph that summarizes the main differences between TNBC with basal vs non-basal like phenotype (in Molecular Classifications of TNBC: Clinical Outcome Implications section pag. 7, lanes 237-253)

-There are some inaccuracies and typos within the text that need to be corrected.

We have reviewed the text and corrected all inaccuracies and typos.